# Assessment of Cost-Effectiveness of Computerized Cranial Tomography in Children with Mild Head Trauma

**DOI:** 10.3390/diagnostics12112649

**Published:** 2022-10-31

**Authors:** Mustafa Çalik, Ayşe Hilal Ersoy, Elif Evrim Ekin, Derya Öztürk, Seda Geylani Güleç

**Affiliations:** 1Department of Emergency Medicine, University of Health Sciences, Gaziosmanpaşa Training and Research Hospital, 34245 Istanbul, Turkey; 2Department of Radiology, University of Health Sciences, Gaziosmanpasa Training and Research Hospital, 34245 Istanbul, Turkey; 3Department of Emergency Medicine, University of Health Sciences, Şişli Hamidiye Etfal Training and Research Hospital, 34371 Istanbul, Turkey; 4Department of Pediatry, University of Health Sciences, Gaziosmanpaşa Training and Research Hospital, 34245 Istanbul, Turkey

**Keywords:** cranial CT, cost-effectiveness, head traumas, malignancy, pediatric

## Abstract

Purpose: Pediatric head traumas constitute the majority of admissions to emergency departments (ED) due to trauma. This study aims to draw attention to the use of cranial computerized tomography (CT) scans in the evaluation of children with head trauma under the age of 18, and to determine CT scans’ usefulness in terms of cost-effectiveness. Materials and Methods: Age, gender, mechanism of trauma and Glasgow Coma Scale (GCS), diagnosis, time of admission to hospital, hospitalization and operation, cranial computerized tomography and hospitalization costs of all cases were retrospectively analyzed. Results: A total of 26,412 patients younger than 18 years old who were admitted to the emergency department due to head trauma and who had a cranial tomography were analyzed. They had a mean age of 7.74 ± 5.66 years. In total, 26,363 (99.8%) of these patients had a GCS greater than 14. Out of these patients, only 402 (1.5%) had brain injury revealed by cranial CT, 41 (0.2%) of these patients were operated and 3 of the patients lost their lives. The total cost of patients admitted to the emergency department with a head injury amounts to USD 583,317. Furthermore, 75.78% of this cost comes from negative cranial CTs. A cost analysis according to different age groups did not show a meaningful difference between 0–2 years and 3–5 years (*p* = 1.000), but there was a meaningful difference for all the other age groups. Conclusion: Our findings show that applying algorithms to predict traumatic brain injury in children with mild head injury rather than scanning all patients with cranial CT will enable more reliable and cost-effective patient care. Current practices should be reviewed to avoid unnecessary radiation exposure and expense in the ED. It is also necessary to inform and educate parents about the risk/benefit ratio of cranial CT scans.

## 1. Introduction

Head injuries in children are a common cause of emergency department (ED) visits. More than 95% of these constitute minor head trauma (MHT), defined as a GCS score greater than or equal to 13. Among all the patients, less than 10% have traumatic brain injuries (TBI) and less than 1% need neurosurgery [1,2]. Traumatic brain injury (TBI) is a major cause of death and disability in children, resulting in more than 7000 deaths, 60,000 hospitalizations and more than 600,000 visits to the ED annually in the US [3,4]. Approximately one-half of these visits involve CT imaging of the head, and the frequency of use of CT has increased substantially in the last decade [5,6]. Considering the limited evidence to guide the acute management of children with minor head injuries, the use of cranial CT scans remains highly controversial. The necessity of using cranial CT scans in the evaluation of children with underdeveloped speaking skills still remains a matter of debate. For example, preverbal children with blunt head trauma are often difficult to assess, and the clinical signs of brain injury may be unclear or overlooked [7].

When compared to plain-film radiography, advanced radiologic imaging such as CT involves much higher doses of radiation and results in a marked increase in radiation exposure. Although our understanding of the carcinogenic potential of low doses of X-ray radiation has increased considerably especially for children, the usage of advanced imaging has also been increasing [8]. Furthermore, other important risks of CT include the need to transport the patient outside the close observation of the ED, risks associated with pharmacological sedation for CT and most importantly the theoretical risk of lethal malignancy from CT. Current estimates of this risk from one cranial CT scan in a child are in the range of 1:2000–1:5000 depending on the age of the child [8,9]. Epidemiological studies show that the radiation dose from even two or three CT scans leads to a detectable increase in cancer risk, especially in children [8].

It is very difficult to assess whether computed tomography is necessary for head injury in children with GCS > 14, especially in infants because they cannot express themselves. This study aims to draw attention to the inessential cranial CT and unnecessary radiation in the evaluation of infants (0–2 years old) and children under 18 years of age, and to determine its usefulness in terms of cost-effectiveness.

## 2. Materials and Methods

This was a retrospective, single-center and cross-sectional study conducted in a local training and research hospital. Our study was conducted in accordance with the Declaration of Helsinki. The local training and research hospital ethics committee approved the study protocol (ethical approval number: 343/10.2021). Since it was a retrospective study, informed consent was not obtained from the patients participating in the studies. We analyzed the admissions to our tertiary ED of the hospital from January 2019 to December 2021. A flowchart of the participants is shown in Figure 1. The patients who were between 0 and 18 years old, admitted to hospital with head trauma and underwent cranial CT scan were included in the study. The exclusion criteria were being older than 18 years old, not having CT imaging, having multiple traumas and being transferred to other hospitals. There are various definitions of periods in a child’s development. In this study, we used the following age-related periods: infant and toddler (1 day–24 months), preschool (2–5 years), school-age child (6–13 years old), and adolescent (13–18 years) [10].

The following data were collected from the hospital database: mechanism of trauma, complaint and GCS at the admission, demographic features, examination findings, neurosurgical consultation report, final diagnosis (International Statistical Classification of Diseases and Related Health Problems 10th [ICD 10] code), clinical follow-up and cost of the overall hospital admission. The Pediatric Glasgow Coma Scale (PGCS) was used to evaluate all children’s groups (smaller than 23 months, between the ages of 2 and 5, bigger than 5 years old) with respect to verbal response, eye opening and motor response [11]. Among the head traumas, traffic accidents and work accidents were reported as judicial cases as well.

We determined the frequency and characteristics of cranial CT scans in the evaluation of children with head trauma. Cranial CT scans were performed with helical multislice CT scanners (GE Optima CT660 128 Slice CT and Siemens SOMATOM Sensation 64 Slice CT) with radiographic slices separated by 5 mm slice thickness. Cranial CT scans were interpreted by radiologists. Abnormal findings detected on cranial CT were defined as follows: signs of diastasis of the skull and/or skull fracture, pneumocephalus, epidural hemorrhage, subdural hemorrhage, subarachnoid hemorrhage, intracranial hemorrhage, contusion, sigmoid sinus thrombosis, traumatic infarction and diffuse axonal injury. Soft tissue trauma, scalp hematoma and cephalohematoma were accepted as normal cranial CT findings. The cost of radiological imaging used in this study was the cost used in all state and training and research hospitals in Turkey and determined by the Ministry of Health. The cost of the trauma was determined in Turkish Lira and converted to United States (US) dollars ($) according to the exchange rates obtained from the data of the central bank to make comparisons and to better understand our results. According to the data of the Central Bank of the Republic of Turkey, the average dollar rate for the years 2019-20-21 was calculated as TRY 5.5, TRY 7 TL and TRY 8.6, respectively, and the average was determined as TRY 6.99. During the study period, the admission and hospital fees were as follows: the CBT scan was between TRY 69.53 and 88.11 (USD 9.93–12.60), the cost of emergency physician service was between TRY 15.50 and 19.60 (USD 2.21–2.80) and the consulting fee from another doctor was between TRY 6 and 7.59 (USD 0.85–1.08) for the Health Implementation Notification (HIN).

### Statistical Analysis

Basic and demographic characteristics were summarized with descriptive statistics. Results were expressed as mean ± standard deviation, range min-maximum and/or a number (percent). The normality distribution of the variables was tested using the Kolmogorov–Smirnov test. Differences between normal and abnormal CT cost groups by age groups were evaluated using the Kruskal–Wallis test and post hoc tests. SPSS 22.0 for Windows (IBM, New York, NY, USA) program was used for the statistical analysis. A *p*-value of <0.05 was considered statistically significant.

## 3. Results

Of the 37,252 patients, 26,412 who met the inclusion criteria were included in the study. There were 9047 (34.3%) female patients and 17,365 (65.7%) male patients. The mean age of the patients was 7.74 ± 5.67 years. The mean age of males was 8.27 ± 5.75 and the mean age of females was 6.72 ± 5.35 years. The male/female ratio was 1.92. Regarding the nationality, 24,258 (91.8%) patients were Turkish citizens, while the remaining 2154 (8.2%) were foreigners. There were 402 (1.5%) abnormal cranial CT scans, and for 773 (2.9%) patients, neurosurgery consultation occurred. In total, 332 (1.3%) patients were hospitalized, 41 (0.2%) patients had an operation and 3 (0.01%) patients died. The demographic and the clinical features of the participants are expressed in Table 1.

Our hospital is a training and research hospital in Istanbul. In the 1065-day period during which we conducted the study, 1,168,018 patients were admitted to our adult ED. Of those, 181,050 patients were followed-up in the emergency trauma room. During this time period, a total of 56,126 cranial CTs were performed in the ED, while 26,412 cranial CTs were performed on pediatric patients with head trauma. Out of the 26,412 patients included in this study, 2008 (7.6%) had multiple head trauma occurring at different time periods. Children with multiple head injuries were reported to Social Services on suspicion of child abuse. The average number of the patients per day who were admitted to the ED with head trauma (0–18 years old) was 35, and approximately 25 of them (71.4%) had cranial CT scans. The time of admission to the ED was similar for both genders, with the busiest admission time between 19:00 and 21:00.

The mechanism of occurrence of head trauma according to the age groups was specified with the ICD 10 code at the time of admission to the ED. In addition, patients were diagnosed with cranial CT. The ICD 10 code and cranial CT diagnoses of the patients according to child age groups are shown in Table 2 in detail.

The total cost of 26,412 patients admitted to the ED with head trauma was USD 583,317. A cost analysis according to different age groups did not show a meaningful difference between 0–2 years and 3–5 years (*p* = 1.000), but there was a meaningful difference for all other groups (*p* < 0.027 between 3–5 and 6–12, and *p* < 0.001 for the remaining age groups). The Kruskal–Wallis test and post hoc tests were used for this comparison. A cost analysis according to the age groups with respect to normal and abnormal cranial CT results is shown in Table 3.

The average cost of patient examination and radiological imaging was USD 22.11 per person. The mean cost in 41 patients who underwent surgery for the diagnosis of head trauma was USD 1669. While the average cost was USD 2179.4 for 49 patients with GCS < 14, it was USD 18.10 for 26,363 patients with GCS ≥ 14. The total cost for 26,010 patients who had normal cranial CT results was USD 442,084. This cost makes 75.78% of all expenditures.

## 4. Discussion

In our study, 71.50% of patients with head trauma under 18 years of age with a GCS ≥ 14 underwent a cranial CT scan. Of these, only 0.8% were hospitalized and 0.06% required surgery. All of the patients with a GCS < 14 were hospitalized, and a surgical operation was performed on 36.7% of them. We found that the severity of trauma increased with age, and applications to the emergency service increased in the evening. Furthermore, we found that school-age children were more exposed to trauma compared to all other age groups, and this group accounted for the majority of tomography costs. We found that the number of judicial attendances such as abuse and beating in the adolescent age group exceeded the number of emergency attendances. In our study, the most common mechanism of head trauma was falling by 55% (23.2% of the patients were 0–2 years). There are different results in other studies about the mechanism of head trauma in children. For example, in the study conducted by Işık et al. which included 851 patients between 0 and 14 years (23% of the patients were 0–2 years), the most common mechanism of trauma was falling by 49.5% (12). In another study conducted by Yıldızhan et al. which included 246 patients with head trauma between 0 and 18 years (15.5% of the patients were 0–2 years), the most common mechanism of trauma was traffic accident by 40.6%, followed by falling by 36.4% (13). Moreover, in these studies it was reported that as the age of the patients increases, the severity of the trauma also increases [12,13]. Our study also supports these results regarding the relation between age and severity of the trauma.

In our study, the male/female ratio was 1.92:1. In recent studies, both in our country and in other countries, male trauma cases (the ratio varying between 2.0:1 and 1.65:1) are dominant, which is similar to our study [14,15]. In the study by Stanley et al. the percentage of the usage of cranial CT scans was analyzed in 25 hospitals with an average number of 26,502 patients per year (range, 10,000 to 89,122) [16]. The study included a total of 26,412 patients admitted to the EDs (<18 years) with head trauma with a GCS of 14–15. For these patients, the usage of cranial CT scan varied between 19.2% and 69.2%. In our study, this rate was 71.5%. In the study of Kuppermann et al. <1% of children with minor blunt head trauma required neurosurgical operation [17]. In our study, similarly, <1% required any neurosurgical operation. It may be useful for excluding the need for non-surgical interventions and neurosurgical intervention. However, we should not be generous in the decision to shoot cranial CT in mild head injuries just to exclude the need for neurosurgery intervention.

Reducing unnecessary radiation from cranial CT is very important in the pediatric population, as studies on radiation exposure, and cancer have estimated the incidence of malignancy following cranial CT in children between 1/1000 and 1/5000, and the rates are estimated to be even higher in the youngest patients [8,18,19]. In addition to the long-term health risks of CT scans, there is another drawback of using CT scans very frequently in terms of costs. Following the above-mentioned incidence of malignancy, we can assume that 5 to 26 children might develop cancer out of 26,412 cranial CTs performed on children in our study. By this assumption, we can estimate the approximate cost of cancer treatment for these patients. There are a few studies on the costs of cancer treatment in Turkey. For example, Bilici et al. reported that the annual average cost of a gastric cancer patient is USD 12,901 [20]. This means that, when we predict that 5–26 patients included in our study might have CT-induced cancer, the annual cost would be USD 64,507––335,438. The five-year survival rate for gastric cancer varies between 5 and 30%. Based on this information, the 5-year cost alone would be between USD 322,535 and 1677,194. In other cancer types, the 5-year survival rate varies between 1 and 99%, and the costs would increase sharply as the survival time increases. In another study, a cost analysis revealed the annual direct medical cost per patient with small-cell lung cancer to be (EUR 8772), USD 10,005, and for non-small-cell lung cancer to be (EUR 10,167) USD 11,590 [21]. These numbers are also similar to the above study, so we can arrive at the same conclusion that the excessive use of cranial CT scans does not only have serious health risks, but also causes a higher cost.

In a retrospective cohort study of 178,604 children in the UK who had at least 1 CT scan (of all types) before the age of 22, at a 10-year follow-up the authors found a positive association between radiation dose from CT scans and leukemia and brain tumors [18]. In our study, recurrent admissions were substantial (7.6%); therefore, we can conjecture a higher risk for these children.

As was mentioned above, in our study, we found that negative CT scan costs made 75.78% of all expenditures, and patients who had negative cranial CT scan constituted 98.5% of the patients. This situation causes an indirect increase in cost by creating a serious waste of effort and time among the personnel working in the hospital: the radiology technicians who perform the scan, and the radiology doctors who evaluate the results.

Computed tomography use in children has doubled in the last 20 years, from 10.6 CT per 1000 children in 1996 to 21.5 per 1000 children in 2010 [22]. However, there are several guidelines for cranial CT indications in children with head trauma, namely the Children’s *Head injury* Algorithm for the prediction of Important Clinical Events (CHALICE) [23], Canadian Assessment of Tomography for Childhood *Head injury* (CATCH) [24], Pediatric Emergency Care Applied Research Network (PECARN) [17] and Scandinavian guidelines for initial management of minor and moderate head trauma in children [25]. We think that effective use of these guidelines in EDs can help in reducing excessive use of cranial CT scans. Another strategy to reduce the use of CT, which is often used in Australia and New Zealand, is to hospitalize patients rather than have an ED CT scan [26]. However, this strategy brings with it the costs and risks of hospitalization. The use of alternative biomarkers can also be helpful in reducing excessive cranial CT scans and possible long-term complications of these scans. In a meta-analysis from recent years, a significant positive correlation was found between the CT scan result and S100B protein in children with mild traumatic brain injury (TBI) [27]. S100B is a well-established traumatic brain injury (TBI) biomarker protein and was added to the Scandinavian guideline for adults, which resulted in a decrease in CT scans by one-third [27]. According to a meta-analysis published by Oris et al. [28] which included 8 eligible studies involving 506 subjects, a significant positive correlation was found between CT scan findings and S100B protein in children with traumatic brain injury (TBI). In that meta-analysis, both S100B testing and CT scan findings showed a pooled sensitivity and specificity, respectively, of 100% (95% CI: 98–100%) and 34% (95% CI: 30–38%). As a result, the authors concluded that S100B protein serum levels, in combination with the PECARN algorithm, could reduce the need for CT scans in children with mild-TBI by one-third. However, a larger dataset to evaluate the effectiveness of other alternatives such as S100B protein and use of transfontanelle ultrasonography (US) in infants compared to cranial CT is required. In addition, studies on the use of transfontanelle US in infants have been carried out in recent years [28,29]. In the study by Trench et al. it was stated that intracranial hemorrhages could be detected with US performed by a trained emergency specialist [28]. In another study conducted with 115 patients, it was reported that the sensitivity of US in detecting skull fracture was 90.9% [30]. More studies are needed on the efficiency of imaging with US for infants, but the use of US, where applicable, could potentially make it possible to avoid cranial CT.

Limitations of this study: the study had several limitations. First, it was limited by the single-center, retrospective design. Second, the records of patients’ reasons for admission to the ED (e.g., nausea, vomiting, headache) could not be accessed through hospital information systems. Third, because pediatric trauma patients were analyzed retrospectively, we could not design certain imaging tools such as transfontanelle ultrasonography (US) to be able to compare its efficacy with cranial CT. Fourth, although not performing cranial CT or another imaging modality (i.e., transfontanelle US in infants) to rule out brain injuries that do not require immediate attention is more cost-effective than scanning all patients with cranial CT, this may present other serious consequences, even in a small number of pediatric TBI patients. In addition, despite the fact that we determined only 402 (1.5%) subjects had a brain injury revealed by cranial CT from a total of 26,412 pediatric head trauma patients admitted to the ED, we cannot draw any conclusions regarding if the cranial CTs are unnecessary or suggesting other non-imaging methods (e.g., measurement of serum S100B and transfontanelle US. These issues should be addressed in future studies involving pediatric patients with head trauma who underwent cranial CT.

## 5. Conclusions

Our physicians in ED who attend to the patients are well-informed about the guidelines for cranial CT scans in children. However, our study indicates that these guidelines are not used effectively to reduce cranial CT scans. We think that one of the reasons for this is due to parents’ lack of awareness about the risks of cranial CT, and them being insistent on having cranial CT scans for their children even after they are informed about the risks by the physicians. We also think that an increase in violence towards healthcare personnel in recent years plays a role in physicians’ decisions to use cranial CT scans on the insistence of parents or relatives of the patients. The emergency physicians’ desire to discharge patients early because of the overcrowded ED might also play a role in using cranial CT scans instead of observing the patients longer. In our opinion, decreasing the use of excessive CT scans can be possible by regular training of the physicians in the ED on this subject, and also by implementing regulations that will enable physicians to perform their professions away from external pressures such as violence, pressure from patients’ relatives or unfair malpractice lawsuits.

## Figures and Tables

**Figure 1 diagnostics-12-02649-f001:**
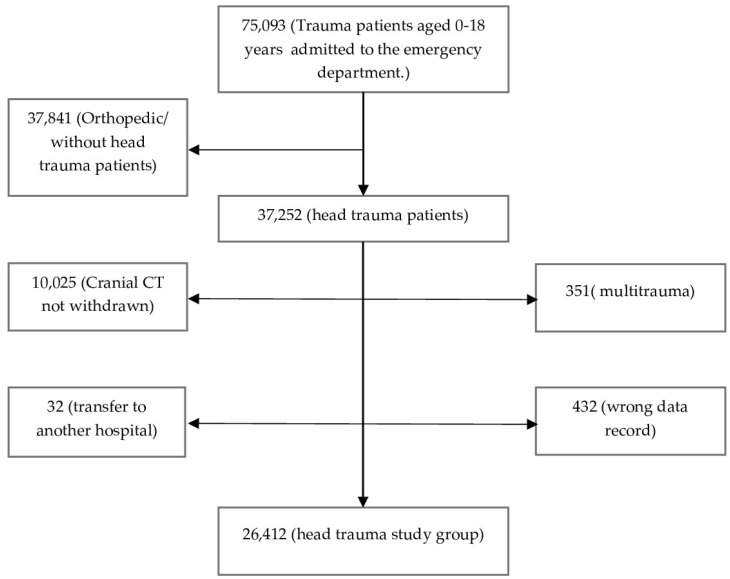
Flowchart of the participants.

**Table 1 diagnostics-12-02649-t001:** The demographic and the clinical features of the participants.

	*n* (%)
**Gender**	
Male	*17,365 (65.7)*
Female	*9043 (34.3)*
Total	*26,412*
**Nationality**	
TR	*24,282 (91.9)*
SY	*1889 (7.2)*
Others	*241 (0.9)*
**Mechanism of trauma**	
Emergency	*21,887(82.9)*
Judicial case	*3091 (11.7)*
Work accident	*124 (0.4)*
Traffic accident	*1310 (5.0)*
**Primary health care provider**	
Specialist	*22,049 (83.5)*
General Practitioner	*4358 (16.5)*
**GCS**	
≥14	26,363 (99.8)
(9–13)	30 (0.12)
<8	19 (0.08)
**Cranial CT Abnormal**	402 (1.5)
**Neurosurgery Consultation**	773 (2.9)
**Hospitalization**	332 (1.3)
**Operation**	**41 (0.2)**

TR; Turkish citizen; SY; Syrian citizen; GCS; Glasgow Coma Scale.

**Table 2 diagnostics-12-02649-t002:** ICD 10 code and cranial CT diagnoses of the patients according to child age groups.

*Variable*	*0–2 Age* *n (%)*	*3–5 Age* *n (%)*	*6–12 Age* *n (%)*	*13–18* *Age n (%)*	*Total n (%)*
	6124 (23.2)	5548 (21.0)	8458 (32.0)	6282 (23.8)	**26,412**
**Diagnosis**					
Soft Tissue Trauma	5834 (95.3)	5452 (98.3)	8370 (99)	6220 (99)	**25,876 (98.1)**
Epidural Hematoma	10 (0.2)	15 (0.3)	11 (0.1)	7 (0.1)	**43 (0.2)**
Subdural Hematoma	4 (0.1)	1 (0)	4 (0)	7 (0.1)	**16 (0.1)**
Subarachnoid Bleeding	9 (0.1)	4 (0.1)	5 (0)	2 (0)	**20 (0.1)**
Intracranial Hemorrhage	3 (0)		2 (0)	4 (0.1)	**9 (0)**
Frontal Fracture	22 (0.4)	16 (0.3)	11 (0.1)	11 (0.2)	**60 (0.2)**
Temporal Fracture	6 (0.1)	9 (0.2)	8 (0.1)	3 (0)	**26 (0.1)**
Occipital Fracture	40 (0.6)	19 (0.3)	15 (0.2)	3 (0)	**77 (0.3)**
Parietal Fracture	44 (0.7)	6 (0.1)	3 (0)	3 (0)	**56 (0.2)**
Multiple Fracture	12 (0.2)	2 (0)	10 (0.1)	3 (0)	**27 (0.1)**
Scalp Hematoma	5 (0.2)	6 (0.1)		6 (0.1)	**17 (0.2)**
Cephalohaematoma	117 (2)				**117 (0.5)**
Contusion	15 (0.1)	18 (0.3)	15 (0.2)	8 (0.1)	**56 (0)**
Pnomocephaly	1 (0)		3 (0)	1 (0)	**5 (0)**
Others	2 (0)		1 (0)	4 (0.1)	**7 (0)**
**ICD_ 10 Code**					
W00-W18-W19	3351 (54.7)	3529 (63.6)	5424 (64.1)	2230 (35.5)	**14,534 (55)**
W50-W51	48 (0.8)	97 (1.7)	395 (4.7)	2193 (34.9)	**2733 (10.3)**
Y28-W25	197 (3.2)	364 (6.6)	730 (8.6)	826 (13.1)	**2117 (8)**
S00-S09	653 (10.7)	637 (11.5)	764 (9.0)	232 (3.7)	**2286 (8.7)**
W22-W23	238 (3.9)	251 (4.5)	511 (6.0)	250 (4)	**1250 (4.7)**
Y30-W06-W10-W14	1601 (26.1)	368 (6.6)	29 (0.3)	480.1)	**2006 (7.6)**
Z04.1	33 (0.5)	298 (5.4)	580 (6.9)	406 (6.5)	**1317 (5.0)**
Z04.2				107 (2.3)	**107 (0.4)**
W34-				21 (0.3)	**21 (0.1)**
V18-	1 (0)	4 (0.1)	24 (0.3)	9 (0.1)	**38 (0.1)**
I46.9	2 (0)		1 (0.1)		**3 (0.0)**

ICD_10: International Statistical Classification of Diseases and Related Health Problems 10th, (W00-W18-W19; Falls, W50–51; hit, struck, kicked, twisted, bitten or scratched by another person, Y28; contact with sharp object, undetermined intent, W25; contact with sharp glass, S00; superficial injury of head, W22-W23; striking against or struck by other objects, caught, crushed, jammed or pinched in or between objects, Y30; falling, jumping or pushed from a high place, undetermined intent, (W06: Fall from bed-W10: Fall on and from and steps-W14: Fall from tree), Z04.1; examination and observation following transport accident, Z04.2; examination and observation following work accidents, W34; discharge from other and unspecified firearms, V18; bicycle rider injury in other and unspecified transport accidents, I46.9; cardiac arrest, unspecified. frequency (*n*) and percentage (%).

**Table 3 diagnostics-12-02649-t003:** Cost-effective analysis with cranial CT according to the age groups of children.

	0–2 Years ^a^	3–5 Years ^b^	6–12 Years ^c^	13–18 Years ^d^	Total	*p*-Value
Cranial CT Abnormal (*n*, %) Group Cost (mean ± SD) * Cost (USD)	168 (0.65%) 85.8 ± 174.3 USD 14,411	90 (0.33%) 92.1 ± 118.4 USD 8285	88 (0.31%) 177.4 ± 343 USD 15,610	56 (0.23%) 1837 ± 6033 USD 102,927	402 (1.52%) 351.3 ± 1785 USD 141,233	* *p* < 0.001 ** *p* ^a–b^ = 1.000 ** *p* ^a–c^ = 0.001 ** *p* ^a–d^ < 0.001 ** *p* ^b–c^ = 0.041 ** *p* ^b–d^ < 0.001 ** *p* ^c–d^ = 0.101
Cranial CT Abnormal (*n*, %) Group Cost (mean ± SD) * Cost (USD)	5956 (22.55%) 16.9 ± 15.6 USD 100,216	5458 (20.67%) 17 ± 11.1 USD 92,968	8370 (31.69%) 17 ± 8.8 USD 142,404	6226 (23.57%) 17.1 ± 8.9 USD 106,496	26,010 (98.48%) 17 ± 19.3 USD 442,084	* *p* < 0.001 ** *p* ^a–b^ = 0.202 ** *p* ^a–c^ < 0.001 ** *p* ^a–d^ < 0.001 ** *p* ^b–c^ = 0.006 ** *p* ^b–d^ < 0.001 ** *p* ^c–d^ < 0.001
Total *n*, (%)	6124 (23.2%)	5548 (21.0%)	8458 (32.0%)	6282 (23.8%)	26,412 (100)	** *p* ^a–b^ = 1.000 ** *p* ^a–c^ = 0.001 ** *p* ^a–d^ < 0.001 ** *p* ^b–c^ = 0.027 ** *p* ^b–d^ < 0.001 ** *p* ^c–d^ < 0.001
* TotalCost (USD)	USD 114,627	USD 101,253	USD 158,014	USD 209,423	USD 583,317	* *p* < 0.001

USD: American Dollars. Variables are presented as mean ± SD, frequency (*n*) and percentage (%). * Kruskal–Wallis Test. ** Post hoc tests, a: 0–2 age groups, b: 3–5 age groups, c: 6–12 age groups, d: 13–18 age groups.

## Data Availability

The authors confirm that the data supporting the findings of this study are available within the article.

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
