# Peer review of "Assessment of Cost-Effectiveness of Computerized Cranial Tomography in Children with Mild Head Trauma"

_diagnostics, 2022, doi:10.3390/diagnostics12112649_

Round 1

Reviewer 1 Report (New Reviewer)

In this retrospective study, the authors investigated the costs related to the use of CT scan in a large sample of pediatric patients admitted to the emergency department of a single hospital for head trauma. The objectives of the study are well-defined, and the study design is appropriate. The methods and results are generally clear, although some clarifications are needed. The discussion is informative, with references relevant to the topic. The conclusions are in general supported by the results, although some sentences might require some clarifications or to be rephrased.

Please find below my specific comments:

1.     “It is very difficult to evaluate head trauma in children with GCS>14 especially in infancy, in terms of the necessity of computed tomography.” This sentence is unclear. Why is it difficult? Please rephrase and clarify.

2.     “However, we estimate that CT is taken in excess.” Is this supported by any reference? Consider also rephrasing "taken in excess".

3.     “The Pediatric Glasgow Coma Scale (PGCS) evaluated children less than 23 months of age, aged 2 to 5 years, and older than 5 years in three parts”. It is not clear if the PGCS was used only for patients less than 2-year-old or for all patients.

4.     In the methods, the authors report “Differences between groups were evaluated using the Student's t-test for normally distributed data and the Mann-Whitney U test for non-normally distributed data. Chi-square test was used to compare the differences of categorical variables between groups. Spearman's Rho correlation was used to determine the relationship between variables.”. However, none of these tests seem to have been used in the study. If so, please remove this part.

5.     In table 1, it is unclear what “Emergency” and “Forensic case” mean in the “mechanism of trauma” section.

6.     “Title( who was examined)” I guess refers to the physician who visited the patient. Please clarify.

7.     In table 2, “Soft Issue Trauma” should be “Soft Tissue Trauma”.

8.     In table 3, it is unclear what the second and fifth rows represent.

9.     Lines 173-176: please clarify which specific variable was analyzed in this comparison and what statistical test was used.

10.   “and applications to the emergency service increases after parents come home from work.” I guess this is a potential interpretation based on the admission time. This should be clarified.

11.   “we found that school-age children are more exposed to trauma”. Compared to which group? Are 3-5-year-old children included in this group? Please clarify.

12.   “We found that the number of judicial at- 193 tendances in the adolescent age group exceeded the number of emergency attendances.” Please clarify what “judicial attendance” means in this context.

13.   “In our study, similarly, <1% required any neurosurgical operation. This suggests that cranial CT scans are overused.” I would argue that the fact that only a minority of patients required a neurosurgical intervention does not necessarily mean that CT scans are overused. In fact, it could still be useful for non-surgical interventions and the exclusion of the need of a neurosurgical intervention.

14.   “Fourth, although, not performing cranial CT or other imaging modality (i.e., transfontanelle US in infants) to rule out brain injuries that does not require immediate attention is more cost-effective than scanning all patients with cranial CT, this may present other serious consequences, even in a small number of pediatric TBI patients.” This does not seem to be a limitation of the study, rather a point to be discussed.

Author Response

Detailed Response to Reviewers’ Comments

Reviewers' comments:

Reviewer#1:
1-“It is very difficult to evaluate head trauma in children with GCS>14 especially in infancy, in terms of the necessity of computed tomography.” This sentence is unclear. Why is it difficult? Please rephrase and clarify.

Answer 1: This statement raised by the reviewer 1 has been rephrased and clarified in the introduction section of the manuscript as: “It is very difficult to assess whether computed tomography is necessary for head injury in children with GCS>14, especially in infants  because they cannot express themselves.” and noted in red bolding.

2-“However, we estimate that CT is taken in excess.” Is this supported by any reference? Consider

Answer 2: The issue raised by the reviewer has been supported in the literatüre (references 5 and 6) as follows: “Approximately one-half of these visits involve CT imaging of the head(5), and the frequency of use of CT has increased substantially in the last decade[5,6].” This ratio is 71.4% in our study, therefore we think it is taken in excess.

Ref. No.5.       National Center for Health Statistics, Centers for Disease Control and Prevention (2000) National Hospital Ambulatory Medical Care Survey, Emergency Department File (2002); CD-ROM Series 13, No. 33

Ref. No.6.       Blackwell CD, Gorelick M, Holmes JF, Bandyopadhyay S, Kuppermann N. Pediatric head trauma: changes in use of computed tomography in emergency departments in the United States over time. Ann Emerg Med. 2007;49(3):320-324. doi:10.1016/j.annemergmed.2006.09.025

3-“The Pediatric Glasgow Coma Scale (PGCS) evaluated children less than 23 months of age, aged 2 to 5 years, and older than 5 years in three parts”. It is not clear if the PGCS was used only for patients less than 2-year-old or for all patients.

Answer 3: The Pediatric Glasgow Coma Scale (PGCS) was used to evaluate all children groups (smaller than 23 months, between the ages of 2 and 5, bigger than 5 years old) with respect to verbal response, eye opening, and motor response. This statement has been clearly stated in lines 97-98 of the manuscript.

4- In the methods, the authors report “Differences between groups were evaluated using the Student's t-test for normally distributed data and the Mann-Whitney U test for non-normally distributed data. Chi-square test was used to compare the differences of categorical variables between groups. Spearman's Rho correlation was used to determine the relationship between variables.”. However, none of these tests seem to have been used in the study. If so, please remove this part.

Answer 4: Thanks for the issue raised by the reviewer. In the statistics section, we have written all the tests we have performed regarding this study, but we have been removed the part of the statistical tests that we did not use in the manuscript. Only the statistics we have been used in the article have been added. We changed this part and noted in red bolding.

5- In table 1, it is unclear what “Emergency” and “Forensic case” mean in the “mechanism of trauma” section.

Answer 5: We havebeen changed the term as “judicial cases” Situations such as abuse, beating, fire-gun injuries have been reported as judicial cases in the hospital records.

6-“Title( who was examined)” I guess refers to the physician who visited the patient. Please clarify.

Answer 6: The term mentioned by the reviewer has been clarified as “Primary health care provider”  which refers both specialist and general practitoner and noted in red bolding in Table 1.

7- In table 2, “Soft Issue Trauma” should be “Soft Tissue Trauma”.

Answer 7: We changed the mentioned term by the reviewer as “Soft Tissue Trauma”.

8- In table 3, it is unclear what the second and fifth rows represent.

Answer 8: All rows have been rearranged properly in Table 3.

9- Lines 173-176: please clarify which specific variable was analyzed in this comparison and what statistical test was used.

Answer 9: In Table 3, p values are indicated in the new table in a more understandable and detailed way. The tests used and the alphabetic letters representing the subgroups are listed under table 3. Lines 173-176 are also written in a more understandable way. The p values of the tests used have been also corrected in the abstract section and noted in red bolding.

10-“and applications to the emergency service increases after parents come home from work.” I guess this is a potential interpretation based on the admission time. This should be clarified.

Answer 10: We have been clarified the paragraph raised by the reviewer as “…applications to the emergency service increases in the evening.”. and noted in red bolding.

11-“we found that school-age children are more exposed to trauma”. Compared to which group? Are 3-5-year-old children included in this group? Please clarify.

Answer 11: “Furthermore, we found that school-age children are more exposed to trauma compared to all other age groups” In lines 89-90 we specify the age groups with respect to ages based on reference number 10. 3-5 years are not included in schoolage children.

12- “We found that the number of judicial at- 193 tendances in the adolescent age group exceeded the number of emergency attendances.” Please clarify what “judicial attendance” means in this context.

Answer 12: We found that the number of judicial attendances like abuse, beating, fire-gun injuries in the adolescent age group exceeded the number of emergency attendances. Situations such as abuse, beating, fire-gun injuries have been reported as judicial cases in the hospital records.

13-“In our study, similarly, <1% required any neurosurgical operation. This suggests that cranial CT scans are overused.” I would argue that the fact that only a minority of patients required a neurosurgical intervention does not necessarily mean that CT scans are overused. In fact, it could still be useful for non-surgical interventions and the exclusion of the need of a neurosurgical intervention.

Answer 13: Thanks for the issue raised by the reviewer. We totally agree with the reviewer's claim. This may not indicate overuse of cranial CT scans. It may be useful for excluding the need for non-surgical interventions and neurosurgical intervention. However, we should not be generous in the decision to shoot cranial CT in mild head injuries just to exclude the need for neurosurgery intervention.

14-“Fourth, although, not performing cranial CT or other imaging modality (i.e., transfontanelle US in infants) to rule out brain injuries that does not require immediate attention is more cost-effective than scanning all patients with cranial CT, this may present other serious consequences, even in a small number of pediatric TBI patients.” This does not seem to be a limitation of the study, rather a point to be discussed.

Answer 14: Thank you for raising this issue. This statement was added previously in the limitation part of the study according to the another reviewer’s suggestion. Furthermore,  discussion section of the manuscript as follows: “In addition, studies on the use of transfontanelle US in infants have been carried out in recent years [29] [30]. In the study by Trench et.al., it was stated that intracranial hemorrhages could be detected with US performed by a trained emergency specialist [29]. In another study conducted with 115 patients, it was reported that sensitivity of US in detecting skull fracture was 90.9% [31]. More studies are needed on the efficiency of imaging with US for infants, but the use of US, where applicable, could potentially make it possible to avoid cranial CT.”

References on the use of Cranial USG on Line 279-283 were included in the discussion section. However, since we do not have data on cranial USG, we could not include it in the discussion part. The study is retrospective and we shown it as a limitation, because the patients are not performed by cranial USG in our emergency department.

Reviewer 2 Report (New Reviewer)

1.       Line 92, adolescent (13-19 years), isn’t the study only including up to 18 years old?

2.       Table 1, “Title( who was examined)”, my understanding is the one performs/orders the procedure. Please use better descriptive phrase

3.       “while 26,412 cranial CT was performed on pediatric patients with head trauma. Out of the 26,412 patients…” Are there anyone who had multiple cranial CT? Are they excluded? Anyone wo had history of cranial CT?

4.       “approximately 25 of them (71.4%) had cranial CT scans” What are the top reasons for them to not get a CT?

5.       Table 3, please arrange the table add “average cost and unit” into the table. P-value is confusing without indication which arms are compared. What are the items that drive the difference between age groups?

6.       Line 215 “In our study, similarly, <1% required any neuro-surgical operation. This suggests that cranial CT scans are overused.” Where are the data to support the claim?

7.       Line 216 “Reducing unnecessary radiation from cranial CT is very important in the pediatric population, as studies on radiation exposure, and cancer have estimated the incidence of malignancy following cranial CT in children between 1/1000 and 1/5000…” The study has nothing to do with the radiation overuse in children, nor has anything to support it.  

8.       Line 227 “when we predict that 5-26 patients included in our study might have CT-induced cancer, the annual cost would be $64,507- $335,438. The five-year survival rate for gastric cancer varies between 5-30%. Based on this information 5-year cost alone would be between $322,535-$1,677,194. In other cancer types, 5-year survival rate varies between 1-99%, and the costs would increase sharply as the survival time increases. In another study cost analysis revealed the annual direct medical cost per patient with small-cell lung cancer to be (€8,772), $10,005 for non-small-cell lung cancer to be (€10,167) $11,590”. This should be analyzed carefully in the result section as cost-effectiveness, instead of casually explained in the discussion section. Authors need data to back-up the assumption in this population

9.       “We also think that increase in violence towards healthcare personnel in recent years plays a role for physicians' decision to use cranial CT scans on the insistence of parents or relatives 304 of the patients.” Could be reasonable but inappropriate to be put into conclusion section without any support

Author Response

Detailed Response to Reviewers’ Comments

Reviewers' comments:

Reviewer #2:

1- Line 92, adolescent (13-19 years), isn’t the study only including up to 18 years old?

Answer 1: Thanks for the issues raised by the reviewer. In our study, all patients were under the age of 18.

2- Table 1, “Title ( who was examined)”, my understanding is the one performs/orders the procedure. Please use better descriptive phrase

Answer 2: The term mentioned by the reviewer has been clarified as “Primary health care provider” and noted in red bolding in Table 1.

3- “while 26,412 cranial CT was performed on pediatric patients with head trauma. Out of the 26,412 patients…” Are there anyone who had multiple cranial CT? Are they excluded? Anyone wo had history of cranial CT?

Answer 3: Of the 26,412 patients included in this study, 2008 (7.6%) had multiple head trauma presenting to the emergency department at different time points.149-150. line was also mentioned.

4-“approximately 25 of them (71.4%) had cranial CT scans” What are the top reasons for them to not get a CT? 

Ansver 4: It is difficult to answer about patients who were out of the study. However It may be that the patient is in a stable condition, the patient's GCS>14 is in good condition, the parents are educated, and the doctor's decision.

5- Table 3, please arrange the table add “average cost and unit” into the table. P-value is confusing without indication which arms are compared. What are the items that drive the difference between age groups?

Answer 5: Thanks for the issues raised by the reviewer. In Table 3, p values are indicated in the new table in a more understandable and detailed way. The tests used and the alphabetic letters representing the subgroups are listed under table 3. Lines 173-176 are also written in a more understandable way. The p values of the tests used were also corrected in the abstract section.

6- Line 215 “In our study, similarly, <1% required any neuro-surgical operation. This suggests that cranial CT scans are overused.” Where are the data to support the claim?

Answer 6: Thanks for the issue raised by the reviewer. We totally agree with the reviewer's claim. We removed that sentence (“This suggests that cranial CT scans are overused.”)This may not indicate overuse of cranial CT scans. It may be useful for excluding the need for non-surgical interventions and neurosurgical intervention. However, we should not be generous in the decision to shoot cranial CT in mild head injuries just to exclude the need for neurosurgery intervention.

7- Line 216 “Reducing unnecessary radiation from cranial CT is very important in the pediatric population, as studies on radiation exposure, and cancer have estimated the incidence of malignancy following cranial CT in children between 1/1000 and 1/5000…” The study has nothing to do with the radiation overuse in children, nor has anything to support it.  

Answer 7: We think that tomography performed in many patients with GKS>14 is unnecessary. Instead, cranial USG or less harmful methods may be recommended. More academic studies are needed on this subject. Our study aims to draw attention to cost of CT scans and possible excessive use of CT. As one of the main drawbacks of CT usage is cancer risk in children we also wanted to draw attention to the cost of cancer treatment which would stem from CT usage. The risks associated with CT scans and cancer in children is rererenced in our study with reference numbers 8, 18, 19 and 22. In addition, the cost of cancer treatment in our country is investigated in other studies, which we gave reference to with numbers 20 and 21. Therefore, our cost estimation on possible future cancer treatment of our dataset is based on previously published studies. We have given the data of those studies and calculated our estimation according to them.

8-   Line 227 “when we predict that 5-26 patients included in our study might have CT-induced cancer, the annual cost would be $64,507- $335,438. The five-year survival rate for gastric cancer varies between 5-30%. Based on this information 5-year cost alone would be between $322,535-$1,677,194. In other cancer types, 5-year survival rate varies between 1-99%, and the costs would increase sharply as the survival time increases. In another study cost analysis revealed the annual direct medical cost per patient with small-cell lung cancer to be (€8,772), $10,005 for non-small-cell lung cancer to be (€10,167) $11,590”. This should be analyzed carefully in the result section as cost-effectiveness, instead of casually explained in the discussion section. Authors need data to back-up the assumption in this population

Answer 8: Thanks for the issue raised by the reviewer. Cost effectiveness has already been studied in our study, and similar study costs have been considered here.

9-“We also think that increase in violence towards healthcare personnel in recent years plays a role for physicians' decision to use cranial CT scans on the insistence of parents or relatives 304 of the patients.” Could be reasonable but inappropriate to be put into conclusion section without any support.

Answer 9: Thanks for the issues raised by the reviewer. “relatives 304 of the patients” I think the number 304 was written by mistake. Because we do not have such data in our article.

Reviewer 3 Report (New Reviewer)

This study refers to the cost-effectiveness of CT scan for pediatric head trauma cases. In pediatric mild head trauma cases, it is not necessary in many cases as mentioned by the author. However, many medical workers would consider the risk of not screening, especially in pediatric cases. For this reason, it would be a better study if you could mention some CT scan specific criteria in these cases.

Author Response

Detailed Response to Reviewers’ Comments

Reviewers' comments:

Reviewer #3:

This study refers to the cost-effectiveness of CT scan for pediatric head trauma cases. In pediatric mild head trauma cases, it is not necessary in many cases as mentioned by the author. However, many medical workers would consider the risk of not screening, especially in pediatric cases. For this reason, it would be a better study if you could mention some CT scan specific criteria in these cases.

Answer: Thanks for the issues raised by the reviewer. It was recommended to follow algorithms which have international validity between lines 244-254 of the manuscript. CT scan-specific criteria are available in the algorithms in reference numbers 17,24,and 25. “However there are several guidelines for cranial CT indications in children with head trauma, namely Children's Head injury Algorithm for the prediction of Important Clinical Events (CHALICE)[24], Canadian Assessment of Tomography for Child-hood Head injury (CATCH)[25], Pediatric Emergency Care Applied Research Net-work (PECARN)[17],”.

Reviewer 4 Report (New Reviewer)

Thanks for the opportunity to review this work. The study has large numbers. It is surprising that so many children had a CT inspite of literature not supporting this for low risk head injury. You have sited parental concern as a possible reason but it would have useful if the reason why the clinician requested for the CT was documented. Are the scans requested by junior medical staff? You have concluded the practice needs to change. Could you elaborate on how you propose to do it. 

Author Response

Detailed Response to Reviewers’ Comments

Reviewers' comments:

Reviewer #4:

Thanks for the opportunity to review this work. The study has large numbers. It is surprising that so many children had a CT inspite of literature not supporting this for low risk head injury. You have sited parental concern as a possible reason but it would have useful if the reason why the clinician requested for the CT was documented. Are the scans requested by junior medical staff? You have concluded the practice needs to change. Could you elaborate on how you propose to do it. 

Answer: Thanks for the issues raised by the reviewer. According to hospital records, the type of primary health care provider was previously stated in Table 1 as “A 22049 (83.5%) of the patients were examined by specialist physicians and 4358 (16.5%) were examined by general practitioners.

Professional experience of specialist physicians ranged from 10 to 20 years, and professional experience of general practitioners ranged from 1 to 5 years.

In the Conclusions;

In addition, we have placed the reason for the excessive use of head CT in patients with mild head trauma in the conclusion section of the manuscript as

“We think that one of the reasons for this is due to parents’ lack of awareness about the risks of cranial CT and them being insistent on having cranial CT scans for their children even after they are informed about the risks by the physicians. We also think that increase in violence towards healthcare personnel in recent years plays a role for physicians' decision to use cranial CT scans on the insistence of parents or relatives of the patients.  The emergency physicians' desire to discharge patients early because of the overcrowded ED might also play a role in using cranial CT scans instead of observing the patients longer.”

Finally, we added following part to the conclusion as well:

“In our opinion, to decrease the use of excessive CT scans can be possible by regular training of the physicians in ED on this subject and also by implementing regulations that will enable physicians to perform their professions away from external pressures such as violence, pressure from patients’ relatives or unfair malpractice lawsuits.”.

Round 2

Reviewer 1 Report (New Reviewer)

The authors satisfactorily addressed all my concerns. I have no further comments.

Reviewer 2 Report (New Reviewer)

Thank the authors for the work. The manuscript tries to address the cost-effectiveness of CT in children with mild head trauma while the studies were insufficient to address the topic. 

Reviewer 3 Report (New Reviewer)

Nothing special to mention.

Reviewer 4 Report (New Reviewer)

Thanks for the revisions. It is interesting to note the pressure from families as a reason for the excess use of CT scans.

This manuscript is a resubmission of an earlier submission. The following is a list of the peer review reports and author responses from that submission.

Round 1

Reviewer 1 Report

The authors perform a retrospective review of children receiving head CTs in the ER and find that the vast majority are negative.  They propose that development of algorithms to avoid unnecessary CT would be more cost effective than scanning all patients.  

The paper has utility in that data on the rate of positive head CT in a peds ER is useful.  

I suggest that the authors delete the speculative remarks about observation and S100B as no data is presented to support these ideas in children. "A longer observation period for the low-risk patients with head trauma in the hospital, use of alternative biomarkers such as S100 B protein etc. can be helpful in reducing unnecessary cranial computerized tomography (CT) scans and possible long-term complications of these scans. "  Further the parental "peace of mind" and system time-savings associated with scanning vs observation is not known.  (Eg does reduced time in the ER enable faster through-put of patients, allowing more rapid recognition and management of true emergencies).  

Author Response

Manuscript Title: Assessment of Cost-effectiveness of Computerized Cranial Tomography in Children with Head Trauma

Journal: diagnostics

Ref. No.: diagnostics-1738347

Reviewers' comments:

Reviewer#1:
The authors perform a retrospective review of children receiving head CTs in the ER and find that the vast majority are negative.  They propose that development of algorithms to avoid unnecessary CT would be more cost effective than scanning all patients. 

The paper has utility in that data on the rate of positive head CT in a peds ER is useful. 

I suggest that the authors delete the speculative remarks about observation and S100B as no data is presented to support these ideas in children. "A longer observation period for the low-risk patients with head trauma in the hospital, use of alternative biomarkers such as S100 B protein etc. can be helpful in reducing unnecessary cranial computerized tomography (CT) scans and possible long-term complications of these scans.

Answer: The statement raised by the reviewer 1 has been omitted from the abstract section of the manuscript. In addition, the new statement mentioned has been addressed in the abstract section of the manuscript as: “Our findings show that applying algorithms to predict traumatic brain injury in children with minor head injury rather than scanning all patients with cranial CT will enable more reliable and cost-effective patient care.” and noted in red bolding.

Furthermore, in the discussion section of the manuscript, we detailed the referenced meta-analaysis to present the data of S 100B protein in children as follows “S100B is a well-established traumatic brain injury (TBI) biomarker protein and is added to the Scandinavian guideline for adults which resulted in a decrease for CT scans by one-third [28]. According to a meta‑analysis published by Oris et al.,[28] which included 8 eligible studies involving 506 subjects, a significant positive correlation was found between CT scan findings and S100B protein in children with traumatic brain injury (TBI). In that meta-analaysis, both S100B testing and CT scan findings showed a pooled sensitivity and specificity respectively of 100% (95% CI: 98%–100%) and 34% (95% CI: 30%–38%). As a result, authors concluded that S 100B protein serum levels, in combination with the PECARN algorithm, could reduce the need for CT scans in children with mild-TBI by one-thirdHowever, a larger dataset to evaluate the effectiveness of other alternatives such as S 100B protein and use of transfontanelle ultrasonography (US) in infants compared to cranial CT is required.” and noted in red bolding.

"Further the parental "peace of mind" and system time-savings associated with scanning vs observation is not known.  (Eg does reduced time in the ER enable faster through-put of patients, allowing more rapid recognition and management of true emergencies). 

Answer: Thanks for the issue raised by the reviewer. The remarks regarding observation in abstract section of the manuscript has been removed as there is no data present to compare this alternative with CT scannig is present as the reviewer rightfully points out.

Reviewer 2 Report

In the manuscript titled ‘Assessment of Cost-effectiveness of Computerized Cranial 2 Tomography in Children with Head Trauma‘, the authors present a retrospective study of the use of CT in pediatric head trauma cases in a hospital in Turkey. The study aims to find whether CT scans are overused, based on whether CT scans detect more serious negative outcomes or the need for surgical intervention.  

My main feedback on the study is that the design is flawed. It is understandable that it’s a retrospective study, which cannot design certain tests to be able to compare efficacy. However, a CT scan is usually conducted in head trauma or TBI patients to be able to rule out serious hemorrhage surrounding or within the brain. A CT-negative patient isn’t automatically safe, and other tests are helpful in defining less severe head trauma. But not having a CT (or other imaging modality) to rule out injuries that require immediate attention may present other serious consequences, even in a small number of patients.

This study has a useful dataset of how often CTs correlate to positive findings, but the claims in the abstract and results are not supported by the presented data. The authors should focus on the main outcomes of the findings and only present the data. Stating that the CTs are unnecessary or suggesting other non-imaging methods (other assessments or blood samples) is not supported by the current data. Studies conducting such assessments (with and without CT) are needed before any conclusions are made. Additionally, a part of the discussion is dedicated to discussing how unnecessary CTs can result in cancer, which carries a higher cost burden. This section also depends on speculative assumptions without the support of the data presented in the manuscript.  

Author Response

Manuscript Title: Assessment of Cost-effectiveness of Computerized Cranial Tomography in Children with Head Trauma

Journal: diagnostics

Ref. No.: diagnostics-1738347

Reviewer #2:

In the manuscript titled ‘Assessment of Cost-effectiveness of Computerized Cranial Tomography in Children with Head Trauma‘, the authors present a retrospective study of the use of CT in pediatric head trauma cases in a hospital in Turkey. The study aims to find whether CT scans are overused, based on whether CT scans detect more serious negative outcomes or the need for surgical intervention. 

My main feedback on the study is that the design is flawed. It is understandable that it’s a retrospective study, which cannot design certain tests to be able to compare efficacy. However, a CT scan is usually conducted in head trauma or TBI patients to be able to rule out serious hemorrhage surrounding or within the brain. A CT-negative patient isn’t automatically safe, and other tests are helpful in defining less severe head trauma. But not having a CT (or other imaging modality) to rule out injuries that require immediate attention may present other serious consequences, even in a small number of patients.

This study has a useful dataset of how often CTs correlate to positive findings, but the claims in the abstract and results are not supported by the presented data. The authors should focus on the main outcomes of the findings and only present the data. Stating that the CTs are unnecessary or suggesting other non-imaging methods (other assessments or blood samples) is not supported by the current data. Studies conducting such assessments (with and without CT) are needed before any conclusions are made. Additionally, a part of the discussion is dedicated to discussing how unnecessary CTs can result in cancer, which carries a higher cost burden. This section also depends on speculative assumptions without the support of the data presented in the manuscript. 

Answer: This issue raised by the reviewer 2 has been placed in the limitation section of the manuscript as: “Limitations of this Study; the study had several limitations. First, it was limited by the single center, retrospective design. Second, the records of patients' reasons for admis-sion to the emergency department (e.g., nausea, vomiting, headache) could not be accessed through hospital information systems. Third, because pediatric trauma patients were an-alyzed retrospectively, we could not design certain imaging tools such as transfontanelle ultrasonography (US) to be able to compare its efficacy with cranial CT. Fourth, although, not performing CT or other imaging modality (i.e., transfontanelle US in infants) to rule out brain injuries that does not require immediate attention is more cost-effective than scanning all patients with cranial CT, this may present other serious consequences, even in a small number of pediatric TBI patients. In addition, despite we determined only 402 (1.5 %) subjects had brain injury revealed by cranial CT from a total of 26,412 pediatric head trauma patients admitted to the emergency department, we cannot draw any con-clusions regarding that the cranial CTs are unnecessary or suggesting other non-imaging methods (e.g., measurement of serum S 100B and transfontanelle US. These issues should be addressed in future studies involving pediatric patients with head trauma who un-derwent cranial CT.” and noted in red bolding.

Furthermore, the statement mentioned by the reviwer 2 regardinghow unnecessary CTs results in cancer, which carries a higher cost burden” has been properly modified without speculative assumptions in the discussion section of the manuscript and noted in red bolding.

Reviewer 3 Report

This manuscript reported the cost-effectiveness of cranial CT (CCT) in children with head trauma. It is well-written, sufficiently presented, and highly suggestive for clinical use. Hereby are some comments/questions:

1. Only 1.5% of patients had CCT lesions, also only 2 % of patients had GCS  <14. It means the sample population belongs to minor trauma. Would it be better to specify the title as ".....in children with mild head trauma"?

2.  The study excluded multi-trauma patients in whom the cost-effectiveness of CCT deserves further analysis or discussion.

3. Were the patients ruled out the possibility of child abuse? Since CCT might be an important clue in surveying this kind of patient.

4. Will it be possible to do the cranial US for young kids less than 1 yr old?   Since the US is a non-invasive tool without the risk of radiation and also gives sufficient information.

Author Response

Manuscript Title: Assessment of Cost-effectiveness of Computerized Cranial Tomography in Children with Head Trauma

Journal: diagnostics

Ref. No.: diagnostics-1738347

Reviewer #3:

This manuscript reported the cost-effectiveness of cranial CT (CCT) in children with head trauma. It is well-written, sufficiently presented, and highly suggestive for clinical use. Hereby are some comments/questions:

  1. Only 1.5% of patients had CCT lesions, also only 2 % of patients had GCS <14. It means the sample population belongs to minor trauma. Would it be better to specify the title as ".in children with mild head trauma"?

Answer 1: The title of the manuscript has rearranged according to the reviewer 3 suggestion as “Assessment of Cost-effectiveness of Computerized Cranial

Tomography in Children with Mild Head Trauma” and noted in red bolding in the title section of the manuscript

  1. The study excluded multi-trauma patients in whom the cost-effectiveness of CCT deserves further analysis or discussion.

Answer 2: Thanks for the issue raised by the reviewer.

A total of 26,412 patients younger than 18 years old who were admitted to the emergency department due to isolated head trauma and who had a cranial tomography were enrolled in this retrospective study. The exclusion criteria were; being older than 18 years old, not having CT imaging, having multiple traumas and being transferred to other hospitals. Therefore, we excluded pediatric head trauma patients who had other injuries. This statement was also presented in the flowchart of the parcipicants.

  1. Were the patients ruled out the possibility of child abuse? Since CCT might be an important clue in surveying this kind of patient.

Answer 3: This issue mentioned by the reviewer 3 has been placed in the results section of the manuscript as: “Children with multiple head injuries were reported to social services on suspicion of child abuse.” and noted in red bolding.

  1. Will it be possible to do the cranial US for young kids less than 1 yr old? Since the US is a non-invasive tool without the risk of radiation and also gives sufficient information.

Answer 4: Thank you for raising this issue. This issue raised by the reviewer 3 has been added in the limitation section of the manuscript as “Because pediatric trauma patients were analyzed retrospectively, we could not design certain imaging tools such as transfontanelle ultrasonography (US) to be able to compare its efficacy with cranial CT.”

However, we have been added recent studies with respect to to transfontanelle ultrasonography (US) in infants, in the discussion section of the manuscript as follows: “In addition, studies on the use of transfontanelle US in infants have been carried out in recent years [29] [30]. In the study by Trench et.al., it was stated that intracranial hemorrhages could be detected with US performed by a trained emergency specialist [29]. In another study conducted with 115 patients, it was reported that sensitivity of US in detecting skull fracture was 90.9% [31]. More studies are needed on the efficiency of imaging with US for infants, but the use of US, where applicable, could potentially make it possible to avoid cranial CT.” and noted in red bolding.